# Improving Inverse Folding for Peptide Design with Diversity-regularized Direct Preference Optimization

## Abstract

Inverse folding models play an important role in structure-based design by predicting amino acid sequences that fold into desired reference structures. Models like ProteinMPNN, a message-passing encoder-decoder model, are trained to reliably produce new sequences from a reference structure. However, when applied to peptides, these models are prone to generating repetitive sequences that do not fold into the reference structure. To address this, we fine-tune ProteinMPNN to produce diverse and structurally consistent peptide sequences via Direct Preference Optimization (DPO). We derive two enhancements to DPO: online diversity regularization and domain-specific priors. Additionally, we develop a new understanding on improving diversity in decoder models. When conditioned on Open-Fold generated structures, our fine-tuned models achieve state-of-the-art structural similarity scores, improving base ProteinMPNN by at least 8%. Compared to standard DPO, our regularized method achieves up to 20% higher sequence diversity with no loss in structural similarity score.

## 1 Introduction

Engineering biopolymers that fold into desired 3D structures, a computational challenge known as inverse protein folding problem, has broad applications in drug discovery and material science (Yang et al., 2023; Dill et al., 2008; Abascal & Regan, 2018). Several approaches for inverse folding have been adopted over the past decades, from molecular dynamics simulations to machine learning approaches (Dauparas et al., 2022b; Shanker et al., 2023; Hsu et al., 2022a; Yi et al., 2023; Correa, 1990). In the standard machine learning approach, a molecular backbone chain serves as input, and a model generates sequences that adopt folding topologies compatible with the reference backbone. Sequences do not necessarily share sequence homology, as multiple diverse sequences can fold into similar structures (Hsu et al., 2022a; Yue & Dill, 1992; Godzik et al., 1993).

Peptides, which are small biopolymers comprising 2-50 residues, are interesting targets for inverse folding given their role in diverse biological functions, acting as hormones, neurotransmitters, signalling molecules, or nanostructures assemblers (Chockalingam et al., 2007; Torres et al., 2019; Copolovici et al., 2014; Ulijn & Smith, 2008). Only about 225,000 protein structures have been experimentally determined[1] and made available via the Protein Data Bank (PDB). Training inverse-folding machine learning models in a supervised fashion is a challenging task, due to the complexity of the problem and the limited amount of experimental data. The challenge is aggravated in the peptide domain as fewer than 3.5% PDB structures contain 50 residues or less. In fact, applying the SCOP classification filter in the PDB to display structures labelled as "Peptide" reveals only 509 entries, circa 0.2% of all experimentally determined structures available.

In addition to the lack of data, sequences are subject to composition bias. The incidence of certain amino acids may differ depending on the sequence length , as longer proteins have more options for accommodating multiple secondary structures and folding loops (Tiessen et al., 2012). Popular models like ProteinMPNN (Dauparas et al., 2022b), PiFold (Gao et al., 2022) and ESM-IF1 (Hsu

---

[1]Updated figures available at https://www.rcsb.org/stats/growth/growth-released-structures.

et al., 2022b) are primarily trained on data derived from longer proteins, leading to poor performance for peptide design tasks. Additionally, shorter sequences fold into simpler structures. Indeed, Milner-White & Russell (2008) argue that short peptides are notoriously "structureless" and tend to flicker between conformations. For example, a structural conformation of a single alpha helix or beta sheet – or a combination of two or three of them – is not necessarily stable and can fluctuate.

Existing inverse folding models optimize sequence residue-recovery accuracy and structural similarity via template modeling (TM) score (Zhang & Skolnick, 2004). However, they often suffer from low sampling diversity (Gao et al., 2023b). Ideally, the inverse folding model generates maximally diverse candidate sequences, as additional design filters, such as synthesizability and thermal stability, reduce the number of potential hits downstream of the design process.

To address these issues, we apply Direct Preference Optimization (DPO) (Rafailov et al., 2023), a fine-tuning method, to improve inverse-folding methods for peptide design. To the authors' knowledge, we are the first to apply DPO to this task. We propose several enhancements to DPO to address specific problems that arise in inverse folding. Particularly, we forward fold generated sequences and derive an online regularized algorithm for optimizing structural similarity to the reference and sequence diversity simultaneously. We show empirically that this algorithm targets the differential entropy in log-probability space. Furthermore, we present a simple reward scaling approach to incorporate scalar reward information, showing that reward scaling adaptively selects a KL divergence penalty (Kullback, S. and Leibler, R. A., 1951) to improve performance on harder structures.

## 2 PRELIMINARIES

**Inverse folding** is the problem of inferring the sequence of amino acids $y$ that fold into a given protein structure. This protein structure is represented by a set of coordinates $x = (x_i, y_i, z_i) \mid i = 1, \ldots, n \in \mathbb{R}^{3n}$, where each $(x_i, y_i, z_i)$ represents the 3D position of a backbone atom and $n$ the number of atoms. The inverse folding problem is underconstrained; there may be many solutions $y$ that fold into structures similar to $x$ (Koehl & Levitt, 1999). Prior research is primarily concerned with recovering the "ground-truth" sequence $y_x$ from the experimental (reference) structure (Dauparas et al., 2022a; Hsu et al., 2022b; Jing et al., 2021b;a). Recently, forward folding models like AlphaFold (Jumper et al., 2021), ESMFold (Lin et al., 2022), and OpenFold (Ahdritz et al., 2022), have made it possible to estimate the structural similarity between generated sequences and the reference structure. In this work, we focus on measuring structural similarity of generated sequences to the reference structure via the self-consistency TM-score (sc-TM) (Gao et al., 2023b).

**ProteinMPNN** (Dauparas et al., 2022a) is a popular inverse-folding method that produces full protein sequences from backbone features (distances and orientations between backbone atoms, backbone dihedral angles, accounting for a virtual C$\beta$ atom). ProteinMPNN is a 6-layer encoder-decoder message-passing neural network based on the 'Structured Transformer' (Ingraham et al., 2019). Unlike autoregressive methods like ESM-IF1 (Hsu et al., 2022b), ProteinMPNN decodes residues in a random decoding order, as opposed to a fixed left-to-right order from N-terminus to the C-terminus. ProteinMPNN is trained on examples from the Protein Data Bank (PDB) to determine the most likely residues for a given protein backbone. On native protein backbones, it achieves a sequence recovery of 52.4%, compared to 32.9% for Rosetta (Leman et al., 2020). In this work we build upon ProteinMPNN by proposing methods for adapting it to peptide design.

**DPO for inverse folding**. Direct Preference Optimization (DPO) is a popular method for aligning Large Language Models to a dataset of human-produced preference assignments, that discriminate amongst the responses grouped by prompt (Rafailov et al., 2023). We adapt DPO to fine-tune inverse-folding models by replacing human preference labels on generated sentences with TM-score rankings on generated sequences. Specifically, we generate a dataset of sequences conditioned on a set of reference structures, and score each sequence with the TM-score computed between its predicted structure and the reference structure. These scores define preference pairs over the generated sequences, for each reference structure.

Let $r(x, y)$ be the TM-score between structure $x$ and the predicted fold of sequence $y$, and $\pi_{\text{ref}}$ be a conditional distribution over $y$ given $x$. Given a dataset $D$ mapping structures ("prompts") to $K$ generated sequences per structure ("responses"), DPO is derived within the KL-constrained reward-maximization objective (Ouyang et al., 2022; Rafailov et al., 2023):

$$\max_{\pi_\theta} \mathbb{E}_{x \sim D, y \sim \pi_\theta(y|x)}[r(x,y)] - \beta \mathbb{D}_{\text{KL}}[\pi_\theta(y \mid x) \mid\mid \pi_{\text{ref}}(y \mid x)] \tag{1}$$

where $\beta$ controls the trade-off between reward maximization and deviation from the base model. The solution to Equation 1 is well-known (Ouyang et al., 2022; Rafailov et al., 2023):

$$\pi_r = \frac{1}{Z(x)} \pi_{\text{ref}}(y \mid x) \exp\left(\frac{1}{\beta} r(x,y)\right) \tag{2}$$

where $Z(x)$ is the partition function to normalize $\pi_r$. Therefore, for any policy $\pi_r$, there is a corresponding reward function (DPO "implicit reward") for which $\pi_r$ is optimal:

$$r(x,y) = \beta \log \frac{\pi_r(y|x)}{\pi_{\text{ref}}(y|x)} + \beta \log Z(x). \tag{3}$$

Substituting Equation 3 into the Bradley-Terry preference model (Bradley & Terry, 1952) and optimizing the policy under a maximum likelihood objective produces the DPO loss, typically solved via gradient descent:

$$\mathcal{L}_{\text{DPO}}(\pi_\theta; \pi_{\text{ref}}) = -\mathbb{E}_{(x,y_w,y_l) \sim \mathcal{D}}\left[\log \sigma \left(\beta \log \frac{\pi_\theta(y_w \mid x)}{\pi_{\text{ref}}(y_w \mid x)} - \beta \log \frac{\pi_\theta(y_l \mid x)}{\pi_{\text{ref}}(y_l \mid x)}\right)\right] \tag{4}$$

where $y_w$ represents the sequence that is preferred or considered better, and $y_l$ represents the sequence that is less preferred or considered worse according to TM-score ranking.

## 3 PREFERENCE OPTIMIZATION FOR PEPTIDE DESIGN

In this section, we consider designing fine-tuning methods well-suited for peptide design. While DPO in its original formulation is useful for fine-tuning in biology (Park et al., 2023; Widatalla et al., 2024; Mistani & Mysore, 2024), we consider how it may be improved to tackle specific problems arising in inverse-folding for peptides (i.e., poor generation diversity and lack of peptide data in initial training). First, we derive a diversity-optimized DPO loss by incorporating an online diversity penalty directly into the top-level Reinforcement Learning objective. Next, we propose an ad-hoc modification to the DPO reward that incorporates scalar TM-scores instead of solely preference pairs, and address the distribution shift between peptides and longer-length proteins.

### 3.1 ONLINE DIVERSITY OPTIMIZATION

To encourage diversity in generated sequences while simultaneously maximizing reward (TM-score), we consider a modified DPO objective incorporating an auxiliary diversity reward:

$$\max_{\pi_\theta} \mathbb{E}_{x \sim D, y \sim \pi_\theta(y|x)}[r(x,y)] - \beta \mathbb{D}_{\text{KL}}[\pi_\theta(y \mid x) \mid\mid \pi_{\text{ref}}(y \mid x)] + \alpha \Gamma_\gamma(\pi_\theta)$$

$$= \max_{\pi_\theta} \mathbb{E}_{x \sim D, y \sim \pi_\theta(y|x)}\left[r(x,y) - \beta \log\left(\frac{\pi_\theta(y \mid x)}{\pi_{\text{ref}}(y \mid x)}\right) - \alpha \mathbb{E}_{y' \sim \pi_\theta(y' \mid x)}[\gamma(y,y')]\right] \tag{5}$$

where $\alpha$ controls the strength of diversity regularization, $\gamma$ is a pairwise distance between sequences, and $\Gamma(\pi_\theta)$ is the diversity of policy $\pi_\theta$ under the distance $\gamma$, i.e. $\Gamma(\pi) = \mathbb{E}_{y,y' \sim \pi}[\gamma(y,y')]$. In practice, we let $\gamma$ be the fraction of pairwise different tokens in equal-length sequences $y$ and $y'$. This is equivalent to the diversity metric defined in (Gao et al., 2023b).

This is similar in form to the auxiliary reward objective proposed by Park et al. (2024) and Zhou et al. (2024), but the auxiliary reward (diversity) depends on the policy $\pi_\theta$. As a result, the standard analytic solution to Eq. 1 (Ziebart et al., 2008; Ouyang et al., 2022; Rafailov et al., 2023) is no longer valid. Instead, we consider an approximate objective, leveraging the fact that the DPO loss is

solved via iterative gradient descent. Let $\tilde{\pi} = \pi_\theta^{(t-1)}$ be an approximation of $\pi_\theta^{(t)}$. That is, we will use the policy from a previous iteration to estimate diversity, while updating the policy in the current iteration. Then, let $\tilde{r}(x,y) = r(x,y) - \alpha\mathbb{E}_{y'\sim\tilde{\pi}(y|x)}\left[\gamma(y,y')\right]$. We can approximate Eq. 5 as:

$$\max_{\pi_\theta} \mathbb{E}_{x\sim D, y\sim\pi_\theta(y|x)}\left[\tilde{r}(x,y) - \beta\log\left(\frac{\pi_\theta(y\mid x)}{\pi_{\text{ref}}(y\mid x)}\right)\right]. \tag{6}$$

The rest of the derivation follows from Rafailov et al. (2023). We produce the diversity-regularized implicit reward after writing the $\tilde{r}$-optimal policy $\pi_{\tilde{r}} = \frac{1}{Z(x)}\pi_{\text{ref}}(y\mid x)\exp\left(\frac{1}{\beta}\tilde{r}(x,y)\right)$:

$$r(x,y) = \beta\log\frac{\pi_r(y|x)}{\pi_{\text{ref}}(y|x)} - \alpha\mathbb{E}_{y'\sim\tilde{\pi}(y'\mid x)}\left[\gamma(y,y')\right] + \beta\log Z(x) \tag{7}$$

The resulting MLE loss under the Bradley-Terry preference model is:

$$\begin{aligned}
\mathcal{L}_{\text{DPO}}\left(\pi_\theta;\pi_{\text{ref}}\right) = -\mathbb{E}_{(x,y_w,y_l)\sim\mathcal{D}}\Big[ &\log\sigma\Big(\beta\log\frac{\pi_\theta\left(y_w\mid x\right)}{\pi_{\text{ref}}\left(y_w\mid x\right)} - \beta\log\frac{\pi_\theta\left(y_l\mid x\right)}{\pi_{\text{ref}}\left(y_l\mid x\right)} \\
&+ \alpha\mathbb{E}_{y'\sim\tilde{\pi}(y'\mid x)}\left[\gamma(y_l,y')\right] \\
&- \alpha\mathbb{E}_{y'\sim\tilde{\pi}(y'\mid x)}\left[\gamma(y_w,y')\right]\Big)\Big]
\end{aligned} \tag{8}$$

The $\alpha$-weighted scalar penalties require sampling from the approximate policy $\tilde{\pi}$ to compute the expectation. In practice, we only update $\tilde{\pi}$ a few times during training to minimize the cost of sampling. See Appendix A.2 for a detailed description of the diversity-regularized algorithm.

Unlike prior methods incorporating auxiliary rewards into DPO (Park et al., 2024; Zhou et al., 2024), ours includes an online sampling term. Empirically, we show that this online term is crucial. In the middle part of Fig. 1, we show that diversity of samples from the static dataset (the auxiliary reward prescribed in an offline approach) does not correlate with the diversity of samples from the trained policy (our auxiliary reward). That is, to accurately estimate – and therefore encourage – diversity over the course of training, online sampling is required. By optimizing the approximate objective in Equation 6, we provide principled motivation for including this online sampling term.

### 3.2 ENTROPY IN DIVERSITY OPTIMIZATION

Here, we consider the mechanism behind diversity regularization, showing that diversity improves due to randomness in the token decoding order during sampling. Particularly, we develop a new understanding of ProteinMPNN based on random decoding order. ProteinMPNN performs random-order token decoding during sampling and all forward passes. So, for a fixed policy $\pi$, sequence $y$, structure $x$, we highlight that $\log\pi(y\mid x)$ is a random variable, depending on a distribution over token decoding orders. Let $\ell_{(\pi,x,y)} = \log\pi(y\mid x)$ denote this random variable, and let $f_\pi(d)$ be a function parameterized by $\pi$ that takes decoding orders $d$ to log-probabilities $s$. In practice, $f_\pi$ is a forward pass through ProteinMPNN, and $d$ is a uniformly chosen permutation over indices $\{1, 2, \ldots, |y|\}$. For a decoding order $d\sim\mathcal{U}$ sampled from the uniform decoding order distribution $\mathcal{U}$, we can compute a log-probability $s = f_\pi(d)$. For this $s$, $\ell_{(\pi,x,y)}(s)$ gives the probability, over all possible $d$, that the log-probability of $y|x$ under $\pi$ is $s$.

**Revisiting training with random log-probabilities**. This new viewpoint allows us to re-evaluate the training of random-order decoder models, like ProteinMPNN. The implicit rewards in Eqs.3 and 7, from which a MLE loss is derived, hide the stochastic nature of $\log\pi(y|x)$. In the explicit decoding order notation, we can write Eq. 3 as

$$r(x,y) = \beta\left[\log\left(f_{\pi_r}(d)\right) - \log\left(f_{\pi_{\text{ref}}}(d)\right)\right] + \beta\log Z(x), \quad d\sim\mathcal{U} \tag{9}$$

with a similar substitution for Eq. 7. This applies to SFT as well, as the loss function has a $\log\pi(y|x)$ term to which we can apply the same substitution. Note that Eq. 9 is computed over a single sample

$d$. That is, training random-order decoders usually involves computing losses with a single-sample estimate of $\log \pi(y|x) = \mathbb{E}_{d \sim \mathcal{U}} \left[ \ell_{(\pi,x,y)}(f_\pi(d)) \right]$. While not explored here, a simple approach to improving the quality of this estimate is to compute log-probabilities over multiple decoding orders, though this would linearly scale training time as well.

**Explaining diversity with differential entropy.** This view also offers insight into how diversity increases without token entropy increasing as well. Within the random log-probability view, there is another source of entropy to account for: the differential entropy $\mathcal{H}_d$ of the distribution over log-probabilities, defined separately for each tuple $(\pi, x, y)$:

$$\mathcal{H}_d(\ell_{(\pi,x,y)}) = \mathbb{E}_{d \sim \mathcal{U}} \left[ -\log \left[ \ell_{(\pi,x,y)}(f_\pi(d)) \right] \right] = -\int_{-\infty}^{0} \ell_{(\pi,x,y)}(s) \log \left[ \ell_{(\pi,x,y)}(s) \right] ds \quad (10)$$

We claim that optimizing for diversity increases differential entropy in the continuous log-probability space, rather than increasing discrete entropy in token space. A proof sketch can be found in Appendix A.6, and empirical support for this theory is presented in Section 4.4.

### 3.3 LEVERAGING DOMAIN-SPECIFIC PRIORS

**Aligning train set and base model.** Supervised fine-tuning (SFT) is a standard part of the DPO pipeline (Rafailov et al., 2023),where a gradient-based optimizer maximizes the log-probability of a target sequence prior to applying DPO. The left part of Fig. 1 shows that the distribution of tokens sampled from base ProteinMPNN and the peptide training dataset differs significantly. SFT assuages the token distribution problem by improving alignment with training distribution, and ensures consistency with previous research, where DPO has been applied to related biological tasks (Park et al., 2023; Widatalla et al., 2024; Mistani & Mysore, 2024).

**Incorporating scalar rewards.** Given multiple responses from a single prompt, DPO is derived assuming access to pairwise preferences. However, since we measure the reward of a sequence generation by the TM-score between the original structure and the generated sequence's predicted structure, we have access to a total ordering over generations through scalar rewards for each response. Widatalla et al. (2024) derives weighted DPO to incorporate scalar rewards, but it does not substantially outperform standard DPO.

We consider a simple ad-hoc method to incorporate these scalar scores. Given a structure $x$, $K$ generated sequences $y_k \,|\, x$, and $K$ corresponding TM-scores $r_k \,|\, x$, we scale the log-probabilities of $\pi_{\text{ref}}$ by the average of $r_k \,|\, x$. To see the effect of this scaling, consider the modified implicit reward:

$$
\begin{aligned}
r_{\text{scale}}(x, y) &= \beta \left[ \log \pi_\theta(y \,|\, x) - R(x) \cdot \log \pi_{\text{ref}}(y \,|\, x) \right] + \beta \log Z(x) \\
&= \beta R(x) \left[ \log \pi_\theta(y \,|\, x) - \log \pi_{\text{ref}}(y \,|\, x) \right] + (\beta - \beta R(x)) \log \pi_\theta(y|x) + \beta \log Z(x) \\
&= \underbrace{[\beta R(x)] \; r(x, y)}_{\text{Reweighted standard reward}} + \underbrace{[\beta - \beta R(x)] \log \pi_\theta(y \,|\, x)}_{\text{Maximum entropy bonus}} + \underbrace{\beta \log Z(x)}_{\text{Partition function}} .
\end{aligned}
\quad (11)
$$

When $R(x)$ is large, the first term (mirroring the standard DPO reward, Eq. 3) has larger weight, and the second term (a penalty on the entropy $\mathcal{H}(\pi_\theta) = -\mathbb{E} \left[ \log \pi_\theta(y \,|\, x) \right]$) has smaller weight. Since the weight on the DPO reward controls the strength of KL divergence regularization (Rafailov et al., 2023), large $R(x)$ corresponds to less aggressive optimization and a lower-entropy policy.

When $R(x)$ is near 1.0, the data-generating policy (i.e., base ProteinMPNN) already produces high-quality sequences on structure $x$. Therefore, we perform less aggressive optimization when $R(x)$ is large, and vice versa. Effectively, per structure, this reward scaling adapts the KL divergence penalty to the strength of the dataset-generating policy. The right part of Fig. 1 shows that this reward scaling method indeed helps ProteinMPNN accurately recover the total ordering over generated sequences. Scaling log-probabilities improves their ability to rank sequences by TM-score, which provides a more accurate reference model for DPO to start with.

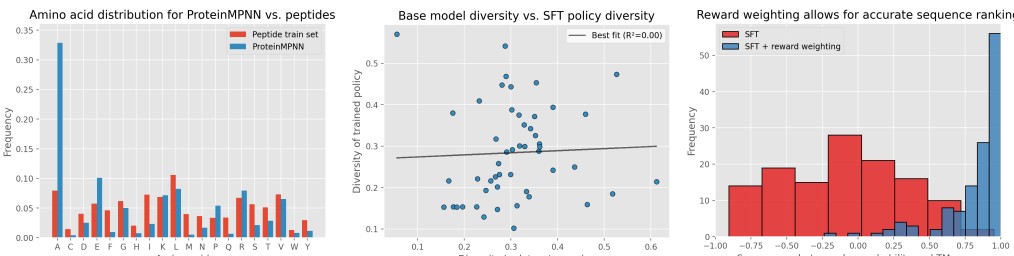

Figure 1: Motivation for DPO design choices. **Left.** Frequency of amino acid across ProteinMPNN generations conditioned on the peptide train set, vs. frequency over the peptide sequences. **Middle.** When conditioned on the same structure, the diversity of sequences generated by base ProteinMPNN does not correlate with the diversity of sequences generated by fine-tuned ProteinMPNN. **Right.** Distribution of rank correlation coefficient between model log-probabilities and TM-score.

## 4 RESULTS

Here we present results on benchmarks across a suite of inverse folding models evaluated on peptide design tasks, as well as an exploration into the behavior of the two proposed algorithm enhancements (diversity regularization and reward scaling). We find that diversity regularization is effective in improving sampling diversity, and provide justification as to how this improvement happens. Additionally, we find reward scaling produces a small improvement in TM-score, enabling fine-tuned ProteinMPNN to reach SOTA performance in some situations.

### 4.1 EXPERIMENTAL OVERVIEW

**Datasets.** We fine-tune trained ProteinMPNN from Dauparas et al. (2022a) on a set of 211,402 deduplicated peptide structures with length up to 50 amino acids, derived from the ColabFold database (Mirdita et al., 2022) by filtering for predicted local distance difference test (pLDDT) $\geq 80$. Each structure was used to generate 4 candidate sequences using pretrained ProteinMPNN with $T = 0.1$. Each sequence, including the true reference sequence from the structure, was folded with OpenFold (Ahdritz et al., 2022). TM-scores were computed to create a ranking over generated sequences for every structure prompt. Each structure therefore contributed $\binom{4}{2} = 6$ chosen-rejected pairs for DPO, for a total of 1,268,412 training pairs. Details of the folding process can be found in Appendix A.1.

**Benchmarks.** We consider two non-overlapping benchmarks. First, we take 50 sequences from the OpenFold set, enforcing a sequence identity threshold of $0.4$ from the train set and filtering for sequence length $L \leq 50$. Next, we take the CATH 4.3 test split, as used in Hsu et al. (2022b), filter for sequence identity threshold of 0.4 and $L \leq 60$, and use the resulting 173 peptides as a second benchmark. The OpenFold set contains structures resolved via OpenFold, while the CATH split has structures from the PDB. For evaluation, we sample 4 sequences per benchmark structure at $T = 0$. We compute diversity, TM-score, and recovery metrics as in Gao et al. (2023b). Following Section 3.1, we define diversity as the average fraction of non-identical amino acids, computed pairwise across all sampled sequences for the same structure.

**Hyperparameters.** Supervised fine-tuning was run for 2 epochs with Adam through the PyTorch implementation (Paszke et al., 2019; Kingma & Ba, 2014), with structures' true sequence as the target. DPO was run for 20 epochs with default Adam hyperparameters on pairs of generated sequences. We vary $\beta$ based on the method used to facilitate a fair comparison across similar KL budgets. For more details, see Appendix $A.1$.

**Sweeps.** We run DPO with diversity penalties $\alpha = \{0.0, 0.1, 0.2, 0.5\}$ and DPO with/without reward scaling, all on the same training set. All training is done with per-GPU batch size of 32 on a single node with 8xA100 80GB GPUs. Each run takes about 8 hours wall-clock time.

### 4.2 BENCHMARK SWEEPS

We benchmark our trained models against standard inverse-folding methods (Jing et al., 2021a;b; Hsu et al., 2022b; Dauparas et al., 2022a) across both the OpenFold and CATH benchmarks, filtered

for peptide-length proteins. We consider TM-score to be the most important metric, as unlike sequence recovery, it can more accurately reflect the quality of generated sequences at high diversities. As shown in Table 1, on OpenFold structures, all DPO methods outperform base ProteinMPNN and other models by at least 8%. Diversity-regularized DPO ($\alpha = 0.1$) achieves state-of-the-art (SOTA) diversity, improving base DPO by 20% and even exceeding the diversity of base ProteinMPNN. This is notable, as fine-tuning tend to decrease generation diversity (Wang et al., 2023). Combining reward scaling with diversity regularization does not have a strong beneficial effect, with similar performance compared to the regularized method.

Table 1: Comparison of models on TM-score, sampled sequence diversity, and native sequence recovery for the OpenFold benchmark. The best results are bolded, and the second-best results are underlined. Means and standard errors are reported in each cell.

| Models | OpenFold benchmark ($n = 50$) | | |
|---|---|---|---|
| | TM-Score ↑ | Diversity ↑ | Recovery ↑ |
| GVP-GNN (Jing et al., 2021b) | $0.62 \pm 0.02$ | $0.19 \pm 0.01$ | $0.27 \pm 0.01$ |
| ProteinMPNN (Dauparas et al., 2022a) | $0.62 \pm 0.01$ | $\underline{0.31 \pm 0.01}$ | $0.23 \pm 0.01$ |
| ESM-IF1 (Hsu et al., 2022b) | $0.61 \pm 0.01$ | $0.00 \pm 0.00$ | $\underline{0.31 \pm 0.01}$ |
| ProteinMPNN + DPO | $\underline{0.66 \pm 0.02}$ | $0.27 \pm 0.01$ | $\mathbf{0.33 \pm 0.01}$ |
| ProteinMPNN + DPO (scaled) | $\mathbf{0.67 \pm 0.02}$ | $0.28 \pm 0.01$ | $0.31 \pm 0.01$ |
| ProteinMPNN + DPO ($\alpha = 0.1$) | $\mathbf{0.67 \pm 0.02}$ | $\mathbf{0.32 \pm 0.01}$ | $0.31 \pm 0.01$ |
| ProteinMPNN + DPO (scaled, $\alpha = 0.1$) | $\mathbf{0.67 \pm 0.02}$ | $\underline{0.31 \pm 0.01}$ | $\underline{0.32 \pm 0.01}$ |

On the CATH benchmark, DPO does not help much (Table 2). This is likely due to the fact that our training method leveraged OpenFold structure predictions, while the CATH benchmark contains experimentally resolved protein structures. However, diversity regularization and reward scaling allow fine-tuned ProteinMPNN to nearly reach the performance of ESM-IF1, which was trained on experimentally resolved structures and is a much larger model. Diversity regularization is still effective in promoting diversity, beating base ProteinMPNN and improving standard DPO by 7%.

Table 2: Comparison of models on TM-score, sampled sequence diversity, and native sequence recovery for the CATH 4.3 benchmark. The best results are bolded, and the second-best results are underlined. Means and standard errors are reported in each cell. Since all values are rounded to 2 decimals, standard errors are not actually zero.

| Models | CATH 4.3 benchmark ($n = 173$) | | |
|---|---|---|---|
| | TM-Score ↑ | Diversity ↑ | Recovery ↑ |
| GVP-GNN (Jing et al., 2021b) | $0.69 \pm 0.01$ | $0.21 \pm 0.00$ | $\mathbf{0.35 \pm 0.01}$ |
| ProteinMPNN (Dauparas et al., 2022a) | $0.68 \pm 0.01$ | $\underline{0.30 \pm 0.00}$ | $0.32 \pm 0.01$ |
| ESM-IF1 (Hsu et al., 2022b) | $\mathbf{0.72 \pm 0.01}$ | $0.00 \pm 0.00$ | $\underline{0.34 \pm 0.01}$ |
| ProteinMPNN + DPO | $0.70 \pm 0.01$ | $0.28 \pm 0.00$ | $0.32 \pm 0.01$ |
| ProteinMPNN + DPO (scaled) | $\underline{0.71 \pm 0.01}$ | $0.29 \pm 0.00$ | $0.32 \pm 0.01$ |
| ProteinMPNN + DPO ($\alpha = 0.1$) | $\underline{0.70 \pm 0.01}$ | $\mathbf{0.31 \pm 0.01}$ | $0.32 \pm 0.01$ |
| ProteinMPNN + DPO (scaled, $\alpha = 0.1$) | $\underline{0.71 \pm 0.01}$ | $\underline{0.30 \pm 0.00}$ | $0.32 \pm 0.01$ |

We find that both reward scaling and diversity regularization are effective in improving TM-score over base DPO, achieving SOTA performance on the OpenFold benchmark. Diversity regularization is particularly effective in improving diversity across both benchmarks. Despite CATH structures being out-of-distribution from the train set, given that they were not produced by OpenFold, our DPO enhancements allow ProteinMPNN to approach the performance of ESM-IF1.

### 4.3 PARETO FRONT WITH DIVERSITY REGULARIZATION

In this section, we consider the impact of diversity regularization on DPO. We train four DPO models on top of fine-tuned ProteinMPNN ($\alpha = \{0.0, 0.1, 0.2, 0.5\}$), and generate sequences across a range

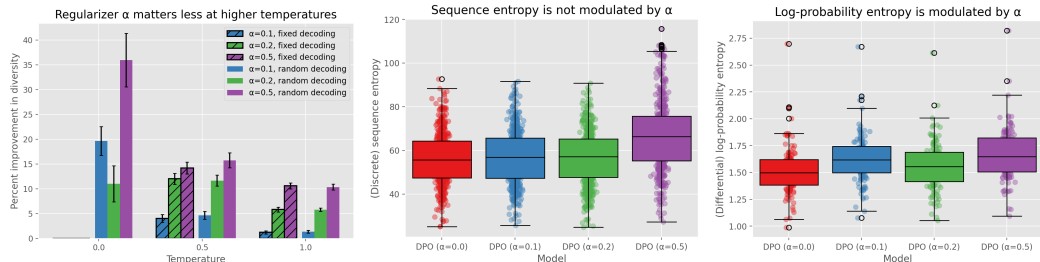

Figure 2: Exploring the effect of online diversity optimization. **Left.** Improvement in diversity across temperatures 0, 0.5, and 1.0, with and without random decoding order. **Middle.** Sampling distribution entropy (average negative log-probability over samples) over various $\alpha$ values. **Right.** Entropy of log-probability distribution over validation reference sequences across $\alpha$ sweep.

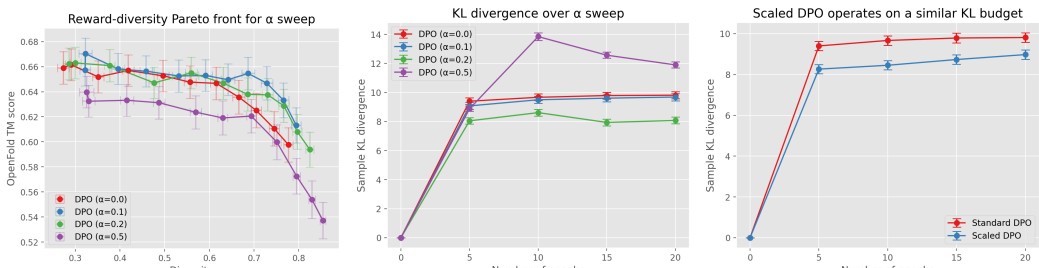

Figure 3: Pareto front and KL divergences. **Left.** Pareto front for various $\alpha$ values over temperature sweep. **Middle.** KL divergence from $\pi_{\text{ref}}$ for various $\alpha$ values. $\alpha = 0$ has $\beta = 0.5$, all other have $\beta = 0.1$. **Right.** KL divergence for DPO with reward scaling ($\beta = 0.1$) and without ($\beta = 0.5$).

of temperatures on the OpenFold test split. As temperature increases, the diversity of generated sequences increases, but the average TM-score (reward) decreases. We consider this reward-diversity tradeoff on the left side of Figure 3. At sufficiently low $\alpha$ values, diversity-regularized DPO produces a new Pareto frontier. With $\alpha = 0.1$, regularized DPO consistently achieves higher reward at the same diversity, indicating this regularization provides a strictly favorable reward-diversity tradeoff compared to simply increasing temperature. At temperature 0.0, $\alpha = 0.1$ yields a 20% relative improvement in diversity along with a small increase in TM-score (1.5%) over standard DPO.

However, at high $\alpha$ values, diversity regularization hurts both diversity and reward. We see symptoms of this in the KL divergences between trained DPO policies and initial fine-tuned Protein-MPNN (middle of Fig. 3). While for $\alpha = \{0.0, 0.1, 0.2\}$, KL divergence trajectories are similar during training, DPO with $\alpha = 0.5$ produces much higher divergences (i.e., the trained policy deviates significantly more from base ProteinMPNN). In this case, aggressive diversity optimization led to a collapse in model capacity via excessive deviation from the initial policy.

### 4.4 DIVERSITY REGULARIZATION TARGETS DIFFERENTIAL ENTROPY

We now consider the explanation for diversity improvement presented in Section 3.2. Naively, we would expect diversity regularization to directly increase the entropy of the sampling distribution $\mathcal{H}(\pi) = -\mathbb{E}_{x \sim \mathcal{D}, y \sim \pi} \left[ \log \pi(y|x) \right]$. In the middle part of Fig. 2, we show that this is not the case. Apart from $\beta = 0.5$ (an outlier as discussed in Section 4.3), increasing diversity does not increase entropy, which seems contradictory. However, this is expected under the differential entropy formulation in Section 3.2, which argued that diversity optimization increases differential entropy in the continuous log-probability space. To test this, we draw 128 samples from $\ell_{(\pi, x, y)}$ for all $(x, y)$ structure-sequence test pairs, i.e., compute log-probabilities with 128 different decoding orders. In the right part of Fig. 2, we show that the estimated $\mathcal{H}_d$ increases with larger $\alpha$, as expected under this theory. This agrees with our intuition, since $\mathcal{H}_d$ controls variability in model log-probabilities, which decide the next token during sampling. That is, larger $\mathcal{H}_d$ leads to greater log-probability variability, leading to greater diversity.

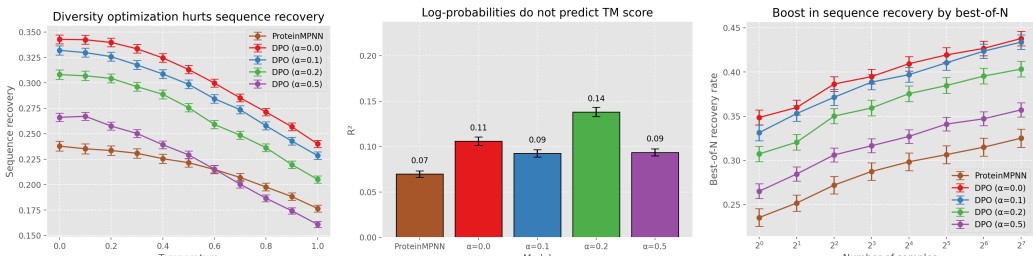

Figure 4: Sequence recovery with diversity optimization. **Left.** Stronger diversity regularization seems to hurt sequence recovery, though all fine-tuned models improve over ProteinMPNN. **Middle.** Diversity does not hurt the correlation between log-probabilities and TM-score. **Right.** Best-of-N sampling allows diverse models to achieve sequence recoveries comparable to standard models.

**Effects at higher temperatures.** In the left half of Fig. 2, we conduct a small ablation study to isolate the effect of random decoding order across temperatures. We find that at low temperatures, random decoding order is necessary for diversity. This is because the next token is chosen deterministically at $T = 0$, so entropy in the log-probabilities is the sole contributor to diversity. However, at higher temperatures, the influence of using random decoding orders is less pronounced, i.e., temperature has a larger effect on next-token sampling compared to an increasing $\mathcal{H}_d$. Therefore, this analysis applies only for random decoding order models at $T = 0$. For example, left-to-right autoregressive methods have fixed decoding orders, meaning $\ell_{(\pi,y,x)}$ collapses around a single scalar and the differential entropy is zero. At $T = 0$, this means that we cannot improve diversity by boosting $\mathcal{H}_d$, and at higher $T$, $\mathcal{H}_d$ does not empirically affect diversity much anyways.

**Increased token entropy does not help diversity.** We also try optimizing for discrete token-level entropy, and achieve around a 15% increase in this entropy. However, in line with the differential entropy formulation, this method does not improve sample diversity at temperature 0. See Appendix A.3 for more details and a complete derivation of the token-entropy regularized algorithm.

### 4.5 SEQUENCE RECOVERY WITH DIVERSITY-REGULARIZED MODELS

In this section, we consider how diversity optimization affects ProteinMPNN's ability to recover native sequences from structures, and accurately predict the quality of generated sequences.

First, we explore whether sequence recovery (i.e., the ability to recover the conditioning structure's sequence) is still possible with diversity-regularized fine-tuning. Intuitively, it may seem that models with higher sampling diversity have lower native sequence recovery rate. In the left half of Fig. 4, it seems like this is the case, with sequence recovery rates decreasing as $\alpha$ increases across all temperatures. However, for small alpha ($\alpha = 0.1$), the drop in sequence recovery compared to standard DPO is small. Additionally, this recovery gap disappears if we allow for more compute during inference time. In the right side of Fig. 4, we take $N$ samples from each model and compute the best-of-$N$ sequence recovery rate. As $N$ grows, the gap between standard DPO recovery rate and $\alpha = 0.1$ recovery rate shrinks to nearly zero. Therefore, optimizing for diversity does not significantly impact sequence recovery rate, particularly as more sequences are sampled.

Gao et al. (2023b) claims that log-probabilities may correlate with sequence quality, e.g. TM-score. Given that diversity regularization increases log-probability entropy as shown in Section 4.4, one might expect this correlation to be less strong under more diverse models. As shown in the middle of Fig. 4, this is not the case. Indeed, across all models, log-probabilities do not correlate with TM-score.

### 4.6 REWARD SCALING REINFORCES BIOLOGICAL PRIORS

Next, we consider the effect of reward scaling on DPO's behavior. On the left side of Fig. 5, the reward-diversity curves for DPO and reward-scaled DPO are computed across $T = \{0.0, 0.1, 0.2, \ldots, 1.0\}$. Reward-scaled DPO achieves consistently higher reward at the same diversity, indicating it is a Pareto improvement over standard DPO. Moreover, as in the right side of Fig. 3, reward-scaled DPO operates on a slightly smaller KL divergence budget compared to

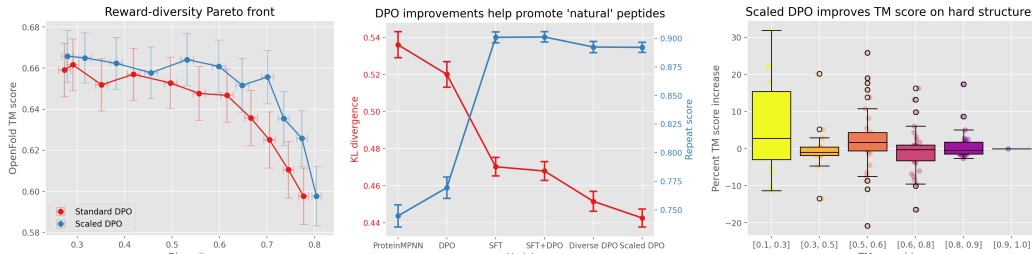

Figure 5: Reward scaling improves DPO. **Left.** Reward-scaled DPO is a Pareto improvement over standard DPO over a temperature sweep. **Middle.** Left axis is the KL divergence between the token frequencies in the peptide train set vs. model samples (lower is better), right axis is the fraction of non-repeating tokens (higher is better). **Right.** TM-score improvement over base DPO. Lower TM-score buckets contain structures for which base ProteinMPNN generated low-quality sequences.

standard DPO. Despite maintaining a smaller deviation from pretrained ProteinMPNN, the policy trained with reward-scaled DPO is still strictly better than the policy trained with standard DPO.

The motivation for reward scaling, as presented in Eq. 11, is dynamic $\beta$ selection based on the strength of the initial policy (or equivalently, the difficulty of the prompt), where $\beta$ controls the KL divergence from the reference model. On the right side of Fig. 5, we show the empirical effect of reward scaling matches this intuition. For structure prompts whose ProteinMPNN-generated sequences had low reward (i.e., where the base model performed poorly), reward-scaled DPO outperforms standard DPO by around 5%. However, as the average TM-score of the generated sequences increases, the performance gain from reward scaling drops to nearly 0. Therefore, reward-scaled DPO targets hard examples where the data-generating policy is weak, agreeing with the motivation in Section 3.3.

## 5 LIMITATIONS

While we illustrate an empirical connection between diversity optimization and differential entropy, we do not establish a mathematical framework for how this optimization happens. Furthermore, during the derivation of diversity-regularized DPO, we approximate $\pi^\star$ with the latest iteration of gradient descent. Exploring the bounds on this approximation, and establishing a theory-first perspective for the differential entropy framework, are promising directions for future research.

Since DPO and the variants proposed here are model-agnostic, they can be used to fine-tune any inverse-folding model. Applying DPO to other inverse-folding models may produce even better results compared to ProteinMPNN Gao et al. (2023b). For example, it may be possible to further push the frontier of peptide sequence design by fine-tuning stronger base models like PiFold (Gao et al., 2022) or KW-Design (Gao et al., 2023a).

## 6 CONCLUSION

We fine-tuned ProteinMPNN, a widely adopted inverse folding model, for diverse and structurally consistent peptide sequence generation and proposed diversity-regularized DPO with an online sampling term to accurately estimate and encourage diversity. Domain-specific priors were also incorporated into our methodology to account for peptides' residue distribution and dynamically control the strength of KL divergence regularization. Our approach results in improvements on sampling diversity, sequence recovery and structural similarity of the generated peptide sequences. Furthermore, we give additional intuition on the impact of diversity regularization on differential and discrete entropy. While our results are reported using ProteinMPNN as a base model for fine-tuning, our proposed methods are agnostic to the inverse folding model, setting the grounds for future research in peptide design via fine-tuning.

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

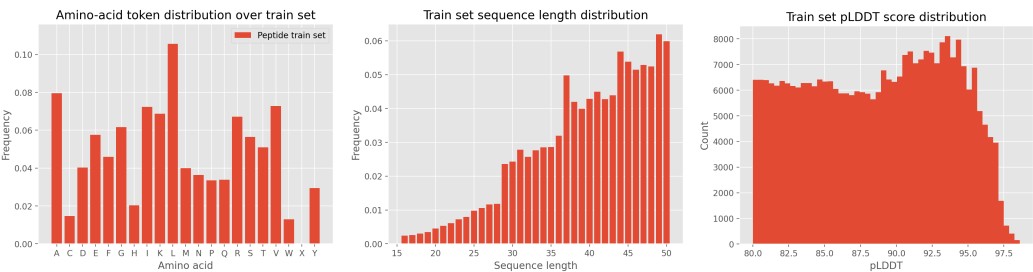

Figure 6: Length, token, and pLDDT statistics on the peptide training set from ColabFold. **Left.** Distribution of amino acid tokens on the train set. **Middle.** Length of sequences in train set. **Right.** Histogram of pLDDT scores from ColabFold. This is after filtering for pLDDT > 80.

# A  APPENDIX

## A.1  ADDITIONAL EXPERIMENTAL DETAILS

**Folding peptide sequences.** Folding of peptides was done with the OpenFold module in NVIDIA BioNeMo Framework (John et al., 2024), version 1.8, with default settings. We used `mmseqs2`(Steinegger & Soeding, 2017) to generate multiple sequence alignments (MSAs), referencing against UniRef90, Small BFD and MGnify datasets, as the input. For template searches, `hhsearch` was used with the PDB70 database. OpenFold inferencing was performed with a single set of weights, converted from an AlphaFold2 (Jumper et al., 2021) model checkpoint `params-model-4`; typically AlphaFold2 is run with five checkpoints. Folding was run on 8 to 32 NVIDIA A100-SXM4-80GB GPUs, with the overall folding throughput around 1.4 seconds per sequence per GPU. We did not perform structural relaxation after folding. For model checkpoint download scripts, and database download scripts, see `github.com/aqlaboratory/openfold`.

**Choosing $\beta$ fairly**. Since $\beta$ is a proxy for specifying the amount of allowable deviation from the base (reference) policy (Rafailov et al., 2023), we ensure fair comparison between DPO and its variants by modifying $\beta$ so that all methods operate on a similar same KL divergence budget. For the diversity-regularized and reward-scaled methods, we choose $\beta = 0.1$; for base DPO, we choose $\beta = 0.5$. In the middle and left parts of Fig. 3, we show that these choices allow all models to deviate from the base policy by around the same amount, with standard DPO still dominating the other model's KL divergences. DPO with $\alpha = 0.5$ is an exception, but we consider this model to be an outlier as described in Section 4.3.

**Method evaluations.** While recent inverse-folding methods like PiFold and KW-Design show strong performance compared to ProteinMPNN (Gao et al., 2023b) and ESM-IF1 Hsu et al. (2022b), we were unable to get their implementation working on the necessary timeline. As a result, we did not include them in our benchmarks.

**Train dataset statistics**. In Fig. 6, we report the distribution of protein lengths, the token-level histograms, and the AlphaFold pLDDT scores for the train dataset referenced earlier.

## A.2  DETAILS OF ONLINE DIVERSITY REGULARIZATION

In Algorithm 1 we present the full online diversity-regularized DPO algorithm. Note that $\gamma$ computes the pairwise diversity between a sequence $y$ and $N$ other sequences $y'$, so it returns an $N$-length vector. $\tilde{\pi}$ is updated only once every $K$ epochs, so in practice we only have to sample once every $K$ epochs. Between these updates, samples are cached and take up only a few megabytes of GPU memory. The modifications compared to base DPO (Rafailov et al., 2023) are highlighted in blue.

## A.3  ENTROPY-REGULARIZED DPO

Here, we consider optimizing for discrete entropy over the sequences sampled from ProteinMPNN. The derivation is the same as in Section 3.1, but with the entropy $\mathcal{H}(\pi) = -\mathbb{E}_{y \sim \pi}[\log \pi(y)]$ as the diversity penalty instead of $\Gamma(\pi)$. The objective is:

---

**Algorithm 1** Diversity-regularized DPO

---

**Require:** Dataset $D = \{(x^{(i)}, y_w^{(i)}, y_l^{(i)})\}$, base policy $\pi_{\text{ref}}$, KL deviation penalty $\beta$, diversity incentive $\alpha$, max steps $N$, sample frequency $K$, sequence distance $\gamma$, number of samples $M$

$\pi_\theta^{(0)} \leftarrow \pi_{\text{ref}}$

$\tilde{\pi} \leftarrow \pi_{\text{ref}}$

**for** $t = 1$ to $N$ **do**

    **if** $t \mod K = 0$ **then**

        $\tilde{\pi} \leftarrow \pi_\theta^{(t-1)}$

    **end if**

    $(x, y_w, y_l) \leftarrow \text{Minibatch}(D)$          $\triangleright$ Batch size $B$, sequence length $L$

    $r(y_w, x) \leftarrow \beta \log \frac{\pi_\theta^{(t-1)}(y_w|x)}{\pi_{\text{ref}}(y_w|x)}$          $\triangleright$ Chosen reward

    $r(y_l, x) \leftarrow \beta \log \frac{\pi_\theta^{(t-1)}(y_l|x)}{\pi_{\text{ref}}(y_l|x)}$          $\triangleright$ Rejected reward

    $s \leftarrow \text{Sample}(\tilde{\pi}, x, M)$          $\triangleright$ Shape $(B, M, L)$

    $d(\tilde{\pi}, y_w, x) \leftarrow \text{Average}(\gamma(y_w, s))$          $\triangleright$ Shape $(B, )$

    $d(\tilde{\pi}, y_l, x) \leftarrow \text{Average}(\gamma(y_l, s))$          $\triangleright$ Shape $(B, )$

    $\pi_\theta^{(t)} \leftarrow \underset{\theta}{\arg\min} -\mathbb{E}_{(x,y_w,y_l)\sim\mathcal{D}} \left[\log \sigma\left(r(y_w, x) - r(y_l, x) + \alpha d(\tilde{\pi}, y_w, x) - \alpha d(\tilde{\pi}, y_l, x)\right)\right]$

**end for**

---

$$\max_{\pi_\theta} \mathbb{E}_{x\sim D, y\sim\pi_\theta(y|x)}[r(x, y)] - \beta\mathbb{D}_{\text{KL}}[\pi_\theta(y \mid x) \;||\; \pi_{\text{ref}}(y \mid x)] + \alpha\mathcal{H}(\pi_\theta)$$

$$= \max_{\pi_\theta} \mathbb{E}_{x\sim D, y\sim\pi_\theta(y|x)} \left[r(x, y) - \beta\log\left(\frac{\pi_\theta(y \mid x)}{\pi_{\text{ref}}(y \mid x)}\right) - \alpha\log\pi_\theta(y \mid x)\right] \tag{12}$$

With the same approximation $\tilde{\pi}$ and modified reward $\tilde{r}(x, y) = r(x, y) - \alpha\log\tilde{\pi}(y \mid x)$:

$$\mathcal{L}_{\text{DivPO}}(\pi_\theta; \pi_{\text{ref}}) = -\mathbb{E}_{(x,y_w,y_l)\sim\mathcal{D}} \Big[\log\sigma\Big(\beta\log\frac{\pi_\theta(y_w \mid x)}{\pi_{\text{ref}}(y_w \mid x)} - \beta\log\frac{\pi_\theta(y_l \mid x)}{\pi_{\text{ref}}(y_l \mid x)} +$$

$$\alpha\log\tilde{\pi}(y_l \mid x) - \alpha\log\tilde{\pi}(y_w \mid x)\Big)\Big] \tag{13}$$

---

**Algorithm 2** Entropy-regularized DPO

---

**Require:** Dataset $D = \{(x^{(i)}, y_w^{(i)}, y_l^{(i)})\}$, base policy $\pi_{\text{ref}}$, hyperparameters $\beta, \alpha$, max steps $N$, $\tilde{\pi}$ update frequency $K$

$\pi_\theta^{(0)} \leftarrow \pi_{\text{ref}}$

$\tilde{\pi} \leftarrow \pi_{\text{ref}}$

**for** $t = 1$ to $N$ **do**

    **if** $t \mod K = 0$ **then**

        $\tilde{\pi} \leftarrow \pi_\theta^{(t-1)}$

    **end if**

    $(x, y_w, y_l) \leftarrow \text{Minibatch}(D)$

    $r(y_w, x) \leftarrow \beta \log \frac{\pi_\theta^{(t-1)}(y_w|x)}{\pi_{\text{ref}}(y_w|x)}$          $\triangleright$ Chosen reward

    $r(y_l, x) \leftarrow \beta \log \frac{\pi_\theta^{(t-1)}(y_l|x)}{\pi_{\text{ref}}(y_l|x)}$          $\triangleright$ Rejected reward

    $\hat{H}(y_w, x) \leftarrow -\alpha\log\tilde{\pi}(y_w \mid x)$          $\triangleright$ No gradient computation

    $\hat{H}(y_l, x) \leftarrow -\alpha\log\tilde{\pi}(y_l \mid x)$          $\triangleright$ No gradient computation

    $\pi_\theta^{(t)} \leftarrow \underset{\theta}{\arg\min} -\mathbb{E}_{(x,y_w,y_l)\sim\mathcal{D}} \left[\log\sigma\left(r(y_w, x) - r(y_l, x) + \hat{H}(y_w, x) - \hat{H}(y_l, x)\right)\right]$

**end for**

---

The full algorithm is described in Algorithm 2, with deviations from base DPO (Rafailov et al., 2023) highlighted in blue. We find that this algorithm does not produce diversity gains (Table 4) or TM-score gains (Table 3). After entropy regularization, sample entropy does increase by around 25% compared to standard DPO; however, diversity does not improve at temperature 0. This supports the differential entropy theory from Section 4.4, since explicitly increasing discrete entropy does not help improve diversity.

Interestingly, as shown in Table 4, at higher temperatures, entropy-regularized DPO does slightly increase diversity. This again aligns with the analysis in Section 4.4, as at higher temperatures, the randomness introduced by the token sampling distribution dominates the randomness introduced by log-probability variations due to random decoding orders.

Table 3: Comparison of TM-Score for ProteinMPNN, standard DPO, and DPO with entropy regularization across different temperatures (T). Same evaluation setting as the OpenFold benchmark.

| Method | TM-Score | | | | | | | | | |
|---|---|---|---|---|---|---|---|---|---|---|
| | T=0.0 | T=0.1 | T=0.2 | T=0.3 | T=0.4 | T=0.5 | T=0.6 | T=0.7 | T=0.8 | T=0.9 |
| ProteinMPNN | 0.62 | 0.61 | 0.62 | 0.61 | 0.62 | 0.61 | 0.60 | 0.57 | 0.56 | 0.56 |
| Standard DPO | 0.66 | 0.66 | 0.65 | 0.66 | 0.65 | 0.65 | 0.65 | 0.64 | 0.63 | 0.61 |
| DPO ($\alpha = 0.1$) | 0.66 | 0.67 | 0.66 | 0.66 | 0.65 | 0.64 | 0.63 | 0.63 | 0.61 | 0.60 |

Table 4: Comparison of diversity for ProteinMPNN, standard DPO, and DPO with entropy regularization across different temperatures (T). Same evaluation setting as the OpenFold benchmark.

| Method | Diversity | | | | | | | | | |
|---|---|---|---|---|---|---|---|---|---|---|
| | T=0.0 | T=0.1 | T=0.2 | T=0.3 | T=0.4 | T=0.5 | T=0.6 | T=0.7 | T=0.8 | T=0.9 |
| ProteinMPNN | 0.31 | 0.35 | 0.39 | 0.47 | 0.54 | 0.59 | 0.65 | 0.71 | 0.74 | 0.78 |
| Standard DPO | 0.27 | 0.29 | 0.35 | 0.42 | 0.50 | 0.56 | 0.62 | 0.67 | 0.71 | 0.75 |
| DPO ($\alpha = 0.1$) | 0.26 | 0.30 | 0.38 | 0.46 | 0.53 | 0.60 | 0.65 | 0.70 | 0.73 | 0.77 |

### A.4 ADDITIONAL BENCHMARK RESULTS

In Tables 5 and 6, we report the same benchmark results as in Tables 2 and 1, but with $N = 64$ samples per structure, as well as at $T = 0.1$. Only ESM-IF and ProteinMPNN results are reported.

Table 5: Comparison of methods on the OpenFold benchmark.

| Method | Score | Diversity | Recovery |
|---|---|---|---|
| ESM-IF1 (Hsu et al. (2022a)) | $0.616 \pm 0.004$ | $0.252 \pm 0.002$ | $\mathbf{0.305} \pm 0.002$ |
| ProteinMPNN (Jing et al. (2021b)) | $0.552 \pm 0.004$ | $\underline{0.421} \pm 0.002$ | $0.155 \pm 0.002$ |
| ProteinMPNN + DPO (scaled) | $\mathbf{0.633} \pm 0.004$ | $0.404 \pm 0.002$ | $0.192 \pm 0.002$ |
| ProteinMPNN + DPO ($\alpha = 0.1$) | $\underline{0.631} \pm 0.004$ | $\mathbf{0.440} \pm 0.002$ | $0.194 \pm 0.002$ |
| ProteinMPNN + DPO | $0.625 \pm 0.004$ | $0.389 \pm 0.002$ | $\underline{0.212} \pm 0.002$ |

### A.5 SAMPLED SEQUENCE EXAMPLE STRUCTURES

In Figures 8 and 7, we present some sequence samples conditioned on structures from both the OpenFold and CATH 4.3 benchmarks. We select pairs where reward scaled DPO and diversity-regularized DPO outperform base ProteinMPNN for visualization.

### A.6 DIFFERENTIAL ENTROPY AND SAMPLE DIVERSITY

We will prove that increasing differential entropy in the log-probabilities increases the expected pairwise difference in next-token sampling. Assume sampling temperature $T = 0$. We prove this for the slightly weaker case considering only next-token sampling, not over the entire sequence.

Table 6: Comparison of methods on the CATH4.3 benchmark.

| Method | Score | Diversity | Recovery |
|---|---|---|---|
| ESM-IF1 (Hsu et al. (2022a)) | **0.719** ± 0.002 | 0.223 ± 0.001 | **0.341** ± 0.001 |
| ProteinMPNN (Jing et al. (2021b)) | 0.662 ± 0.002 | 0.316 ± 0.001 | 0.318 ± 0.001 |
| ProteinMPNN + DPO | 0.688 ± 0.002 | 0.315 ± 0.001 | 0.317 ± 0.001 |
| ProteinMPNN + DPO (scaled) | 0.701 ± 0.002 | 0.308 ± 0.001 | 0.320 ± 0.001 |
| ProteinMPNN + DPO ($\alpha = 0.1$) | 0.695 ± 0.002 | **0.340** ± 0.001 | 0.317 ± 0.001 |

Suppose $Y_1 \ldots Y_n$ are discrete random variables over the set of amino acid tokens, where $Y_i$ is conditioned on $Y_{j<i}$. This represents the generative process of sampling token $i$ conditioning on the previous sampled tokens. For simplicity, we omit reference to the conditioning backbone structure, the distribution $Y_1$ can be implicitly viewed as capturing this dependence.

Let $y_1 \ldots y_{i-1}$ be a partial realization of $Y_1 \ldots Y_n$ up to the $(i-1)$-th token. Consider two sequences $A$ and $B$, where $A = y_1 \ldots y_{i-1}, y_A$ and $B = y_1 \ldots y_{i-1}, y_B$, where $y_A, y_B$ are independent samples from $Y_i \mid y_1 \ldots y_{i-1}$. We omit the dependence of $Y_i$ on $y_1 \ldots y_{i-1}$ in the notation from here on. Per the definition of diversity in Section 3.1, the pairwise diversity on the partial sequences $\gamma(A, B)$ is dependent only on $\mathbb{1}(y_A \neq y_B)$. Assuming the next token of $A, B$ to be conditionally independent given $y_1 \ldots y_{i-1}$, we consider this quantity in expectation over $Y_i$,

$$
\begin{aligned}
\mathbb{E}[\mathbb{1}(y_A \neq y_B)] &= 1 - P(y_a = y_b) \\
&= 1 - \sum_{y \in S} P(Y_i = y, Y_i = y \mid y_1 \ldots y_{i-1}) \\
&= 1 - \sum_{y \in S} P(Y_i = y \mid y_1 \ldots y_{i-1})^2
\end{aligned}
\tag{14}
$$

where $S$ is the set of amino acid tokens. Maximizing Eq. 14 corresponds to minimizing the sum of squared probabilities, i.e. $\mathbb{E}[P(Y_i)]$. Noting that $\log$ is concave and applying Jensen's inequality, we can derive a lower bound on the diversity via the discrete Shannon entropy $\mathcal{H}(Y_i)$,

$$
\begin{aligned}
\log \mathbb{E}[P(Y_i)] &\geq \sum_{x \in S} P(Y_i = y) \log P(Y_i = y) = -\mathcal{H}(Y_i) \\
\mathbb{E}[P(Y_i)] &\geq \exp\{-\mathcal{H}(Y_i)\} \\
P(y_A \neq y_B) &\geq 1 - \exp\{-\mathcal{H}(Y_i)\}
\end{aligned}
\tag{15}
$$

That is, if the entropy of $Y_i$ increases, so does the expected diversity of the next token [2]. Now, since $T = 0$, $Y_i = \operatorname{argmax}\{\ell_1, \ldots, \ell_{|S|}\}$ where $\ell_k$ is the log-probability of the $k$-th token. In the notation of Section 3.1, $\ell_k = \ell_{(\pi, x, \{y_1 \ldots y_{i-1}, S_k\})}$, i.e. a random variable defined as a transformation over uniform decoding orders given by the parameters of the MPNN.

Since $\ell_k$ are all mutually dependent with negative pairwise correlation (the sum of their exponentials must be one) and share the same support, increasing the differential entropy $\mathcal{H}_d(\ell_k)$ across all all $|S|$ log-probabilities necessarily increases $\mathcal{H}(Y_i)$, as the uncertainties of the individual variables determine the uncertainty of their maximum. Therefore, larger differential entropies in the log-probabilities leads to larger next-token entropies, which lower-bounds the expected next-token diversity in independent samples.

---

[2]Note this is a subtly different type of entropy compared to the one referenced in the middle part of 2. When $T = 0$, as in this proof, discrete sequence entropy is 0, since the expectation is over the non-existent stochasticity in the token sampling process. However, when $T = 0$, the quantity $\mathcal{H}(Y_i)$ is nonzero, since the expectation is over the stochasticity in decoding orders.

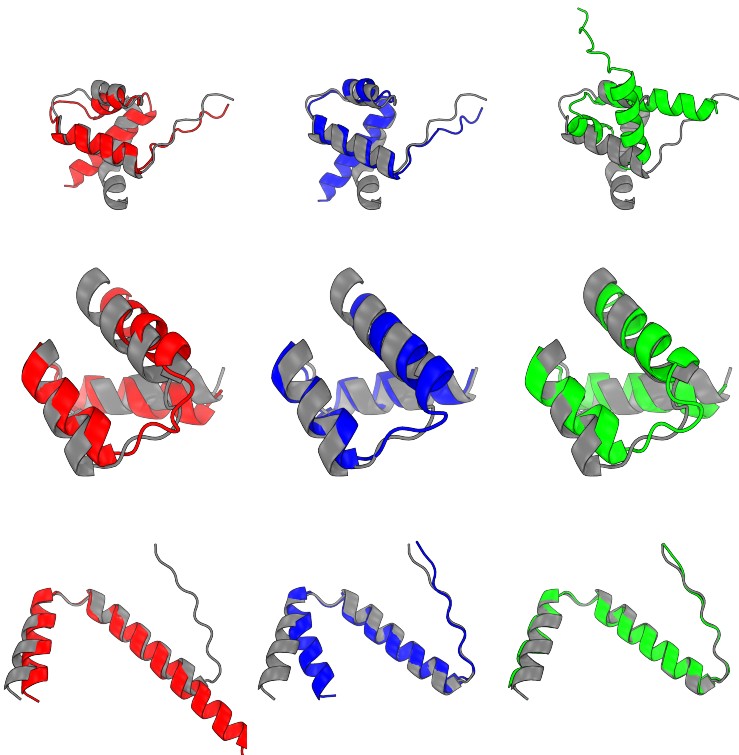

Figure 7: Sequence samples on three randomly chosen structures from the CATH 4.3 peptide benchmark, all folded with OpenFold. Gray is the reference structure (OpenFold), red is a sample from base ProteinMPNN, blue is from diversity-regularized DPO, and green is from reward-scaled DPO.

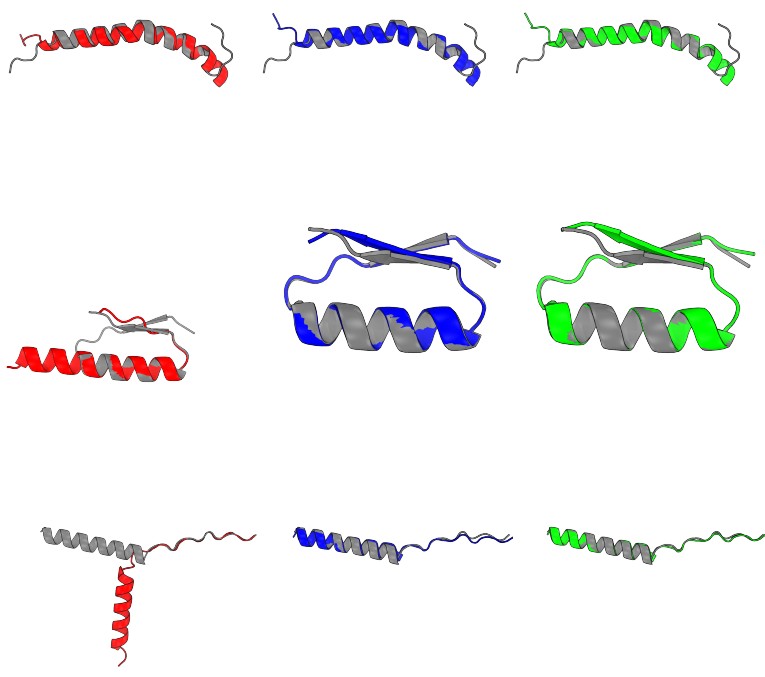

Figure 8: Sequence samples on three randomly chosen structures from the OpenFold peptide benchmark, all folded with OpenFold. Gray is the reference structure (OpenFold), red is a sample from base ProteinMPNN, blue is from diversity-regularized DPO, and green is from reward-scaled DPO.

