# OpenReview forum: "Improving Inverse Folding for Peptide Design with Diversity-Regularized Direct Preference Optimization"
_ICLR.cc/2025/Conference — Submitted to ICLR 2025_

### Official Review · Reviewer_Un8E · 2024-10-16

**Soundness:** 3
**Presentation:** 3
**Contribution:** 2
**Rating:** 3
**Confidence:** 5

**Summary:**

This paper claims two main problems:

The first one is current inverse folding methods cannot generate samples that can be folded to accurate structures sharing high similarity with the input ones (RMSD and TM-Score). The second one is current human verified peptide (<50 AA) structures is limited, leading to poorer performance in the inverse folding models.

The authors propose a novel optimization method to address this problem via DPO while promoting sampling diversity, and obtain State-Of-The-Art performance on peptide deisgn tasks, with the evaluation metrics inculding TM-Score, Diversity, and Recovery.

**Strengths:**

1. The proposed challenges are attractive in this field. The inverse folding samples often suffers low TM-Scores, and diversity. Specially, towards peptide sequenes, the recovery rates are often lower than the samples with other lengths.

2. The motivation behind the method is clear and reasonable. DPO-related methods on auto-regressive-based models work on the field of NLP. Furthermore, the diversity improvement is also impressive.

3. The experiment is comprehensive w.r.t. analysis metrics.

**Weaknesses:**

1. The verfication of the proposed method is limited. There's so many popular approaches on protein inverse folding, such as PiFold, ESM-IF, LM-Design, VFN, and InstructPLM [1-5]. Regarding recovery and self-consistency TM-Score, there are approaches performing better than or competitive than ProteinMPNN. Thus, a robust method with board application should include more experiments to verify the effectiveness of the proposed methods.

2. The proposed metric is not comprehensive enough, the self-consistency RMSD is also required to verifying the quality of sequence samples.

3. The method is simple but the experiement results are not impressive enough to prove the effectiveness of DPO-based improvement. In Table1, the relative improvement over all metrics is smaller than or equal to 1% except TM-Score with 6%. The ablation results cannot indicate the power of proposed method enhancement.

4. The length scale of main experiments are somehow not consistent, which makes the paper unclear to the readers. In Table1&2, the author states that they have had experiments on CATH4.3 (n=173) and Openfold (n=50), but 1) the filtering process is not clear in Openfold benchmark; 2) the filtering criteria of two benchmarks are inconsistent.

Reference:

[1] Gao Z, Tan C, Chacón P, et al. PiFold: Toward effective and efficient protein inverse folding[J]. arXiv preprint arXiv:2209.12643, 2022.

[2] Hsu C, Verkuil R, Liu J, et al. Learning inverse folding from millions of predicted structures[C]//International conference on machine learning. PMLR, 2022: 8946-8970.

[3] Zheng Z, Deng Y, Xue D, et al. Structure-informed language models are protein designers[C]//International conference on machine learning. PMLR, 2023: 42317-42338.

[4] Mao W, Zhu M, Sun Z, et al. De novo protein design using geometric vector field networks[J]. arXiv preprint arXiv:2310.11802, 2023.

[5] Qiu J, Xu J, Hu J, et al. InstructPLM: Aligning Protein Language Models to Follow Protein Structure Instructions[J]. bioRxiv, 2024: 2024.04. 17.589642.

**Questions:**

1. Why you choose different filtering process of two benchmark datasets?
2. Why the diversity of ESM-IF on both datasets is zero?
3. Once the DPO-based method is effective to inverse folding methods, why the experiments are only conducted to the general protein. It is because I believe that the challenges you stated also exist in the general proteins.
4. In the paragraph "Explaining diversity with differential entropy", can you please try to explain the interpretability via a mathematical proof to clearly state the connections between log-probabilities and diversity?

---

> ### Author Response · Authors · 2024-11-27
> **Response to Reviewer Un8E**
>
> Dear Reviewer,
>
> Thank you for your comments and suggestions! We hope our responses can clarify some of your concerns.
>
> > The verfication of the proposed method is limited. There's so many popular approaches on protein inverse folding, such as PiFold, ESM-IF, LM-Design, VFN, and InstructPLM [1-5]... Regarding recovery and self-consistency TM-Score, there are approaches performing better than or competitive than ProteinMPNN. Thus, a robust method with board application should include more experiments to verify the effectiveness of the proposed methods.
>
> We agree with the reviewer’s assessment here: our point is to show that our DPO method is better than no finetuning, as well as the original DPO. Due to some data access issues, we were unfortunately unable to run any other benchmarks besides the ones in Tables 1 and 2. Unfortunately, there’s not much we can do here - we hope the reviewer can understand.
>
> > The proposed metric is not comprehensive enough, the self-consistency RMSD is also required to verifying the quality of sequence samples.
>
> Thank you for this suggestion! We agree and apologize for this oversight. Unfortunately, we were not able to compute these during the review period.
>
> > The method is simple but the experiement results are not impressive enough to prove the effectiveness of DPO-based improvement. In Table1, the relative improvement over all metrics is smaller than or equal to 1% except TM-Score with 6%. The ablation results cannot indicate the power of proposed method enhancement.
>
> We will address each metric separately to show that our DPO method does indeed significantly outperform standard DPO and the base method.
>
> On recovery: the recovery metric is not a well-suited metric, as even if all generated sequences are high quality (high TM-score, or RMSD with target), native sequence recovery is by definition anticorrelated with the diversity of the sequence set. We report it only for consistency with prior work.
>
> On TM-Score: as the reviewer noted, there are clear gains with DPO. These gains, importantly, are not diluted by applying diversity-regularized DPO, which matches the performance of DPO while also jointly optimizing for diversity.
>
> On diversity: we note that applying DPO to ProteinMPNN results in a model with 14% less diverse generations but 8% better TM-Score. However, our DPO method is able to match the original diversity of ProteinMPNN while maintaining that 8% boost in TM-Score. Therefore, while DPO is unable to significantly beat vanilla ProteinMPNN’s diversity, it erases the diversity loss we get from finetuning. That is, with our method, we achieve a Pareto improvement along the diversity-TMScore frontier compared to standard DPO and ProteinMPNN.
>
> Thus, our method achieves the diversity/TM-Score co-optimization objective originally set forth.
>
> > The length scale of main experiments are somehow not consistent, which makes the paper unclear to the readers. In Table1&2, the author states that they have had experiments on CATH4.3 (n=173) and Openfold (n=50), but 1) the filtering process is not clear in Openfold benchmark; 2) the filtering criteria of two benchmarks are inconsistent.
>
> Apologies for the confusion. The OpenFold set was constructed by randomly sampling 128 proteins from the larger train set (from the ColabFold database with sequence length <= 50), then filtering sequences at 0.4 sequence identity threshold level. Since we ran OpenFold experiments first and did not want to modify the filtering criteria for CATH 4.3 since it is standard, we were unable to match the two criteria between benchmarks.
>
> > Why the diversity of ESM-IF on both datasets is zero?
>
> Tables 1 and 2 report results at T=0. Since ESM is fixed decoding order, its sampling diversity is zero at T=0.
>
> > Once the DPO-based method is effective to inverse folding methods, why the experiments are only conducted to the general protein. It is because I believe that the challenges you stated also exist in the general proteins.
>
> The algorithms we designed were selected for peptide design - for example, it is less likely that the diversity problem is as pronounced in general protein design, since two shorter sequences have a greater chance of being similar compared to two longer sequences. However, we agree with the reviewer that our algorithms can be applied to general sequence design for co-optimizing diversity and self-consistency score. We only run experiments on peptide inverse folding here, so we make no claims about other domains (general proteins) but suspect similar results may hold.
>
> > In the paragraph "Explaining diversity with differential entropy", can you please try to explain the interpretability via a mathematical proof to clearly state the connections between log-probabilities and diversity?
>
> Thank you for the suggestion! We add a proof sketch in the appendix.
>
> Thank you again for your patience with our responses! We hope this clarifies your concerns and look forward to additional discussion.

---

> > ### Comment · Reviewer_Un8E · 2024-11-28
> > **Reply to the author's response**
> >
> > Thank you very much for your claim and your concerns!!
> >
> > I fully understand the challenges you are facing. in this stage, I think the algorithm and novelty for the paper is near boarderline, and the experiment design is comprehensive. However, I am still confused about whether the application & experiment scale are enough. It's because peptide design without binding or other constriants with only MPNN's result is not convincing, and the readers cannot still fetch useful conclusions.
> >
> > Actucally, I am very likely to raise score to 4. The reasons are very simple, the method design and the experiment design deserve 6, while the experiment results and scale only deserve 3 or lower in my mind. I am so sorry for your situation for computing resource limitation. In my perspective, AI4Biology paper should have either of experimental results and simple-yet-brilliant novel ideas.
> >
> > In my view, if you can extend the experiment to general protein, using more RL-based methods as baseline (such as PPO), and testing different inverse folding baselines (such as PiFold, VFN, InstructPLM...), it would be a very promising paper! In this stage, I prefer to keep my score fixed. But I am expecting your response and your thoughts about the special contributions of your paper to persuade me.

---

> > > ### Author Response · Authors · 2024-11-29
> > > **Response to Reviewer Un8E**
> > >
> > > Dear Reviewer,
> > >
> > > Thank you for your response and willingness to engage in discussion! We agree that the experiment scale is limited. However, we argue that our algorithm is a meaningful step forward in fine tuning inverse folding models. In particular, diversity-regularized DPO is the first model-agnostic method to improve diversity post-training. It is also the first online DPO approach in protein design, one which we derive/explain in a principled fashion.
> > >
> > > First, we highlight that our diversity-regularized DPO is the first method to apply DPO to diversity optimization to the author’s knowledge. We developed this method after noting that standard DPO, applied to ProteinMPNN, yields 7-14% worse diversity despite an increase in self-consistency score. So the need for our method is evident. With ~10% improvements in diversity and no loss in TM-score over standard DPO, our method is successful in this goal.
> > >
> > > Second, recent research in the DPO literature [2, 3, 4] has outlined the effectiveness of online sampling in DPO for increasing performance downstream. To the author’s knowledge, ours is the first work to consider online DPO in protein design at all (other methods like ProteinDPO and [5] are strictly offline). Furthermore, we produce an online DPO method by a principled derivation from the KL-constrained RL objective, whereas prior DPO work generally applies online sampling ad-hoc without justification [2], [3]. We also provide an empirically supported theory (see the differential entropy formulation, results, and proof) on why online DPO is helpful compared to offline DPO.
> > >
> > > Third, we added a new appendix with evaluation settings more useful to readers (i.e., N=64 samples instead of 4 samples, T=0.1 instead of 0.0). We see similar, if not better gains - on the AF benchmark, for example, our methods achieve around 13% diversity gains over regular DPO, and beat regular ProteinMPNN by 14%. Though we were unable to run a large benchmark suite, our most important comparison is not between SOTA models and ours, since our method can be applied to any differentiable inverse folding model. Our DPO methods beat standard DPO when fine tuning ProteinMPNN in multiple benchmarks. It is simply a matter of implementation to switch out ProteinMPNN for newer, better IF models.
> > >
> > > Lastly, we had considered PPO but decided against it since DPO largely performs similarly, is much easier to train, and takes much less compute [1]. In fact, PPO considers the same KL-constrained objective that DPO is derived from. Mathematically, our online DPO algorithm is a consequence of a simple change to this objective and nothing more - in principle PPO can be modified to incorporate a similar diversity objective in the same vein. Our method is general in this sense since it was produced in a principled manner at the objective level.
> > >
> > > Thank you again for your patience and willingness to discuss! It is greatly appreciated by the authors. To conclude, readers can fetch the useful conclusion that 1) DPO is helpful for diversity optimization, 2) online DPO beats offline DPO, and 3) generally, post-training alignment is helpful for improving inverse folding on particular targets (i.e. peptides).
> > >
> > > [1] Direct Preference Optimization: Your Language Model is Secretly a Reward Model
> > >
> > > [2] Bootstrapping Language Models with DPO Implicit Rewards
> > >
> > > [3] Direct Language Model Alignment from Online AI Feedback
> > >
> > > [4] Understanding the performance gap between online and offline alignment algorithms
> > >
> > > [5] Preference optimization of protein language models as a multi-objective binder design paradigm

---

> > > > ### Comment · Reviewer_Un8E · 2024-11-29
> > > > **Response to the authors**
> > > >
> > > > Thank you very much for your reponse!
> > > >
> > > > I think your insight to your method improvement is fine for me. I understand and agree with your statement.
> > > >
> > > > However, there's still some concerns:
> > > >
> > > > 1. What is application value of increasing the diversity of protein sequence design models. To be concise, I think current models' diversity is somehow enough for design new samples. Actually, the most challenging part for structure-based sequence design model is how to generate sequences that share the same structure with the input sequences, which is evaluated by computing the TMScore/RMSD of the folded structures with original strucutes.
> > > >
> > > > 2. Actually, I can understand the advantages for "DPO" and "Online". I think the theory part is the more effective. But the limited scale of experiments cannot prove the effectiveness of your online DPO in general, which is also consistent with my original concern.
> > > >
> > > > 3. Could you address if peptide design has some special value/challenges compared to general protein design?
> > > >
> > > > Many thanks for your reply again. Look forward to your reply!

---

> > > > > ### Author Response · Authors · 2024-11-29
> > > > > **Response to Reviewer Un8E**
> > > > >
> > > > > Thank you for your response and continued discussion!
> > > > >
> > > > > 1. We argue that inverse folding will continue to benefit from more diverse generations, since it means more fully representing the distribution of matching sequences for a given structure, and is useful for candidate generation tasks in drug discovery pipelines. We also cite recent research in designing more diverse IF models [1, 2, 3] to claim that while progress has been made, simultaneously generating diverse and structure-matching sequences is still quite unsolved.
> > > > >
> > > > > 2. We agree experimental scope is limited in the number of models we benchmark, and do want to highlight our theory/algorithm as the main point of the paper. However, we argue that our benchmarks, which cover 223 experimentally resolved and computationally predicted structures (via CATH 4.3 and AF respectively) across a multitude of protein families, provide sufficient - if non exhaustive - support for our hypothesis. Particularly given that we constrain ourselves to peptides only, and not the general protein case (see bullet 3), we believe readers can draw the conclusion that online DPO for diversity optimization is a worthwhile effort for specialized inverse folding.
> > > > >
> > > > > 3. Yes, peptide design has a number of unique challenges, two main ones being 1) the lack of available peptide data to train on, and 2) the inherent difficulty in modeling shorter proteins. First, fewer than 3.5% of PDB structures contain less than 50 amino acids, making peptide inverse folding more difficult due to lack of data. We have also established in the paper how ProteinMPNN, GVP etc perform worse in the peptide benchmarks than in he general protein setting. This agrees with intuition as peptides exhibit different amino-acid expression levels than longer proteins, as in [5]. Secondly, peptides have a greater tendency to lack rigid/single structure [4], making the inverse folding problem harder to model, and more in need of attempts to recover diverse sequences to deal with this inherent flexibility.
> > > > >
> > > > > We look forward to additional comments and hope this can clarify some doubts.
> > > > >
> > > > > [1] Reinforcement learning on structure-conditioned categorical diffusion for protein inverse folding
> > > > >
> > > > > [2] Graph Denoising Diffusion for Inverse Protein Folding
> > > > >
> > > > > [3] Uncovering sequence diversity from a known protein structure
> > > > >
> > > > > [4] Predicting the conformations of peptides and proteins in early evolution
> > > > >
> > > > > [5] Mathematical modeling and comparison of protein size distribution in different plant, animal, fungal and microbial species reveals a negative correlation between protein size and protein number, thus providing insight into the evolution of proteomes.

---

### Official Review · Reviewer_BXke · 2024-10-19

**Soundness:** 2
**Presentation:** 3
**Contribution:** 2
**Rating:** 5
**Confidence:** 3

**Summary:**

This paper introduces a novel DPO method to finetune inverse folding models on peptide design task, by adding online diversity regularization objective function and incoporating scalar reward into original DPO.

**Strengths:**

1. The concept of employing DPO for fine-tuning peptide inverse folding models is innovative.
2. Leveraging ColabFold and the Openfold database to address the data scarcity issue in peptide entries shows great promise.
3. The online diversity formulation appears to be an improvement over the original diversity-regularized objectives, and the inclusion of differential entropy is particularly inspiring.

**Weaknesses:**

1. As discussed in Sections 3.1 and 3.2, incorporating diversity regularizations and scalar weighted control into DPO is not a novel technique; it has been extensively studied in prior work, which may limit the paper's novelty.
2. The experimental setup is not comprehensive, particularly in terms of baseline details and sampling settings.
3. The presentation in Section 4 is somewhat unconventional; the theoretical analysis in Section 4.4 appears confusing and out of place.

**Questions:**

1. The author should provide more comparisons to prior works, highlighting the differences, such as online optimization or ad-hoc scalar weighting. If possible, please include the original diversity regularization DPO and scalar weighting DPO in your baselines to study the improvements over these methods.
2. The baseline results are limited and may not be convincing.
+ Clarify how the baselines are used: are they pretrained or fine-tuned? Since baseline methods like GVP-GNN are not trained on the OpenFold benchmark, this comparison may not be fair. Please provide clarification.
+ In line 267, the paper claims to use 4 samples and T=0, which is an unusual hyperparameter setting. T=0 implies deterministic sampling of ProteinMPNN (T=0.1 by default), and 4 samples are too few. I encourage using more samples (e.g., 64) and the original T setting.
+ Additionally, I recommend including more baselines, particularly fixed-backbone models specific to peptide design tasks (e.g., PepFlow by Li et al. 2024).
3. In Section 4.4, the paper introduces a new method using entropy instead of pairwise distance as a regularization term. However, this setting is not presented in the main results. Please include this in the main results.

---

> ### Author Response · Authors · 2024-11-27
> **Response to Reviewer BXKe (1/2)**
>
> Dear Reviewer,
>
> Thank you for your time and effort in your review! We appreciate your feedback and hope that our responses can clarify the points you have raised.
>
> > As discussed in Sections 3.1 and 3.2, incorporating diversity regularizations and scalar weighted control into DPO is not a novel technique; it has been extensively studied in prior work, which may limit the paper's novelty… The author should provide more comparisons to prior works, highlighting the differences, such as online optimization or ad-hoc scalar weighting. If possible, please include the original diversity regularization DPO and scalar weighting DPO in your baselines to study the improvements over these methods.
>
> Our apologies for the confusion. To clarify, diversity regularization DPO and scalar weighted DPO are our original work.
>
> To the author’s knowledge, we are the first to use DPO to improve sampling diversity - this is our first contribution. Our second contribution is deriving an online sampling term in the DPO algorithm, and providing justification for why our method works. Recent work [1, 2, 3] in the DPO literature has shown that online DPO, i.e. augmenting the train dataset with online samples, is helpful. However, previous work applying DPO to proteins [4, 5] only consider offline DPO. So, our work is novel in three ways: it considers online DPO for proteins, presents a principled derivation for the online sampling part, and is the first to consider DPO for diversity optimization at all.
>
> Our last contribution is the scalar weighted DPO method, which has intuitive motivation presented in Section 3.2. This is similar to the “weighted DPO” method from ProteinDPO [4] in that both methods leverage continuous rewards instead of preferences as in standard DPO. However, in [4], weighted DPO does not significantly outperform standard DPO; we attempt to do better.
>
> > The presentation in Section 4 is somewhat unconventional; the theoretical analysis in Section 4.4 appears confusing and out of place.
>
> We have updated the rebuttal revision, moving the theoretical analysis to Section 3 in methods.
>
> > The experimental setup is not comprehensive, particularly in terms of baseline details and sampling settings… The baseline results are limited and may not be convincing… Additionally, I recommend including more baselines, particularly fixed-backbone models specific to peptide design tasks (e.g., PepFlow by Li et al. 2024).
>
> We were unfortunately unable to run the full set of benchmarks we wanted. Many newer popular methods are implemented in ProteinInvBench [6], but we were unable to get the benchmark harness to produce reasonable results with existing models, so we did not report the results here. We agree that the argument would be strengthened with additional baselines/experiments, and thank you for the specific direction towards PepFlow. Unfortunately, due to logistical constraints, it is no longer feasible for us to run any additional benchmarks. We hope you can understand in this regard.
>
> > Clarify how the baselines are used: are they pretrained or fine-tuned? Since baseline methods like GVP-GNN are not trained on the OpenFold benchmark, this comparison may not be fair. Please provide clarification.
>
> The benchmark models are pre-trained. We would like to highlight that the most important comparison we are making is between vanilla ProteinMPNN, ProteinMPNN + standard DPO, and ProteinMPNN + our DPO methods. Ideally, we have similar DPO finetuned results for each base model. We unfortunately were not able to implement DPO for the other base models, but report their non-finetuned results for clarity.
>
> However, our results suggest that our DPO methods do indeed outperform ProteinMPNN, as well as standard DPO. In Table 1, vanilla ProteinMPNN has 14% higher diversity than ProteinMPNN + DPO, but ProteinMPNN + DPO has 8% higher TM-Score. ProteinMPNN + our diversity-regularized DPO is able to maintain the diversity level of vanilla ProteinMPNN while also maintaining the 8% higher TM-Score that comes with fine tuning.
>
> Thank you again for your patience with our responses! We hope this clarifies your concerns and look forward to additional discussion.
>
> [1] Bootstrapping Language Models with DPO Implicit Rewards
>
> [2] Direct Language Model Alignment from Online AI Feedback
>
> [3] Understanding the performance gap between online and offline alignment algorithms
>
> [4] Preference optimization of protein language models as a multi-objective binder design paradigm
>
> [5] Aligning protein generative models with experimental fitness via Direct Preference Optimization
>
> [6] https://github.com/A4Bio/ProteinInvBench

---

> ### Author Response · Authors · 2024-11-27
> **Response to Reviewer BXKe (2/2)**
>
> Dear Reviewer,
>
> We respond to two remaining comments. Apologies for the lengthy response.
>
> > In Section 4.4, the paper introduces a new method using entropy instead of pairwise distance as a regularization term. However, this setting is not presented in the main results. Please include this in the main results.
>
> Our apologies again for the confusion. The difference between the entropy optimization algorithm in the appendix and the main diversity-regularization algorithm in Section 3 is that the former is completely offline while the latter is online. In the language of our differential entropy theory, offline entropy optimization naively tries to upweight the discrete entropy of sampled sequences, while online diversity regularization implicitly upweights the differential entropy of log-probabilities themselves. As per the appendix, the offline method fails to meaningfully improve diversity. To clarify, the offline method is our work as well, a simple algorithm whose poor performance supports the differential entropy theory.
>
> Thus, the entropy method is not part of our core results, but rather further justification for our choice in designing an online DPO method. Particularly, it provides empirical support for our theoretical argument about differential entropy optimization in online diversity-regularized DPO.
>
> > In line 267, the paper claims to use 4 samples and T=0, which is an unusual hyperparameter setting. T=0 implies deterministic sampling of ProteinMPNN (T=0.1 by default), and 4 samples are too few. I encourage using more samples (e.g., 64) and the original T setting.
>
> Thank you for the suggestion! We have sampled with 64 instead of 4 samples per structure and at T=0.1, the results are in the appendix of the revised version. The ordering of the results does not change. On the AF benchmark, for example, our methods achieve around 13% diversity gains over regular DPO, and beat regular ProteinMPNN by 14%.

---

### Official Review · Reviewer_9iKQ · 2024-10-20

**Soundness:** 3
**Presentation:** 2
**Contribution:** 3
**Rating:** 6
**Confidence:** 4

**Summary:**

This paper proposes a novel approach that could help design peptide sequences. It mainly uses an advanced DPO structure to finetune ProteinMPNN. The proposed model obtained a significant improvement in various metrics. The paper is easy to follow.

**Strengths:**

- The introduction of DPO into inverse folding is interesting. Finetuning Graph-based models may provide some insights into designing a smaller structural encoder.
- The paper integrates structural information into training (maximizing TM-score) in the inverse folding tasks. It is a nice try but it may also include bias (seen in the next part)
- The design of the reward function and the loss function avoids the reliability of the policy term, it is an interesting approximation!

**Weaknesses:**

- In line 58, the use of Tmscore appears abruptly, as the models listed in the previous paragraph are purely structure-to-sequence models, there needs to be a further explanation about the idea of Inverse Folding, which is actually a structure-sequence-structure pipeline. Although it has been addressed in the Part 2.
- The use of the TM-score in training may lead to bias since the TM-score is obtained through a static structure prediction model. Although it may not be easy to analyze the reason, I think at least the authors can mention that and do some analysis (like using multiple structure prediction tools to calculate the TM-score to average the bias). It might be a little bit time-consuming, so it doesn’t have to be done in the rebuttal phase and it would have no effect on my rating. Just a concern.
- The approach itself is a bit target-specific, it did not work well on the co-optimization purpose as the author claims. From Table 1, the diversity and recovery do not excel from others, I am not blaming the results but curious about the reason.

**Questions:**

- As the paper compared to GVP, ProteinMPNN, and ESMIF, why doesn’t the paper compare with some more recent approaches? Since the paper cited the ProteinInvBench, it provided a model zoom and colab demo to re-run the newer models easily if the input sequence is ready.
- Could there be an appendix for analyzing the statistical features of the dataset used in training?

**Details Of Ethics Concerns:**

No related concerns.

---

> ### Author Response · Authors · 2024-11-27
> **Response to Reviewer 9iKQ**
>
> Dear Reviewer,
>
> Thank you very much for your comments and insights. We are grateful for the opportunity to respond and clarify where possible. Thank you for your patience!
>
> > The use of the TM-score in training may lead to bias since the TM-score is obtained through a static structure prediction model. Although it may not be easy to analyze the reason, I think at least the authors can mention that and do some analysis (like using multiple structure prediction tools to calculate the TM-score to average the bias). It might be a little bit time-consuming, so it doesn’t have to be done in the rebuttal phase and it would have no effect on my rating. Just a concern.
>
> We agree with the reviewer. Our initial plan was to use the 5 trained AF model weights to bootstrap the computation of TM-scores. However, due to infrastructure limitations, we were only able to use 1 such model for structure inference.
>
> > The approach itself is a bit target-specific, it did not work well on the co-optimization purpose as the author claims. From Table 1, the diversity and recovery do not excel from others, I am not blaming the results but curious about the reason.
> Agreed. We report it only for consistency with prior work. We would like to highlight that the recovery metric is not a well-suited metric, as even if all generated sequences are high quality (high TM-score, or RMSD with target), native sequence recovery is by definition anticorrelated with the diversity of the sequence set.
>
> On the diversity metric, we highlight that our methods achieve a Pareto improvement over vanilla ProteinMPNN and ProteinMPNN + standard DPO. Applying standard DPO to ProteinMPNN produces a model with 14% worse diversity but 8% better TM-Score compared to the vanilla (base) model. However, applying our diversity-regularized DPO approach to ProteinMPNN produces a model with slightly better diversity and still 8% better TM-Score compared to the base model. Thus, our method does indeed achieve the diversity/TM-Score co-optimization objective originally set forth.
>
> > As the paper compared to GVP, ProteinMPNN, and ESMIF, why doesn’t the paper compare with some more recent approaches? Since the paper cited the ProteinInvBench, it provided a model zoom and colab demo to re-run the newer models easily if the input sequence is ready.
>
> We were able to set up benchmarks with ProteinInvBench, but due to some unresolved issues in the benchmarking repository, the results we observed with these newer approaches were not sensible. We were ultimately unable to resolve these issues, and so could not in good faith put out comparisons to the models in ProteinInvBench. While ideally we could make these comparisons, we appreciate your understanding in this regard.
>
> > Could there be an appendix for analyzing the statistical features of the dataset used in training?
>
> Thank you for the suggestion. We have added an appendix considering the length distribution, token distribution, and pLDDT score distribution of the train set.
>
> Thank you again for your patience with our responses! We hope this clarifies your concerns and look forward to additional discussion.

---

> > ### Comment · Reviewer_9iKQ · 2024-11-28
> >
> > Thanks for the detailed reply. My questions were largely answered. I will increase my score. However, could the authors confirm what is the problem in the ProteinInvBench? It is usable at least on my machine. I do encourage the authors to compare their approach with those protein design methods.

---

> > > ### Author Response · Authors · 2024-11-28
> > > **Response to Reviewer 9iKQ**
> > >
> > > Thank you for the reconsideration! We were able to set up ProteinInvBench and run the provided code with our benchmarks, but could not get the pretrained models to produce results as expected (ie sampled sequences and their corresponding folds were nonsensical). As a result, we could not in good faith compare our methods without falsely representing the other design methods.
> > >
> > > Thank you for the understanding in this matter, we definitely would have liked to compare with more/newer methods. Time and resource constraints made it difficult to do much debugging with ProteinInvBench. Our main comparison, however, is against the regular DPO algorithm, rather than any specific base model, since our DPO method can be applied to any such base model.

---

> > > > ### Comment · Reviewer_9iKQ · 2024-11-28
> > > >
> > > > Understood, thanks for the explanation!

---

### Official Review · Reviewer_sGWi · 2024-10-29

**Soundness:** 3
**Presentation:** 3
**Contribution:** 2
**Rating:** 5
**Confidence:** 5

**Summary:**

This paper studies an important problem in protein design, inverse folding. The authors propose to fine-tune ProteinMPNN with DPO to generate diverse and structurally consistent peptide sequences. In experiments, the fine-tuned ProteinMPNN achieves state-of-the-art structural similarity scores and diversity.

**Strengths:**

- The paper is well-written and easy to follow.
- The authors observe that popular inverse folding models perform poorly for peptide design.
- This paper applies DPO to improve diversity.

**Weaknesses:**

- ProteinDPO [1] has applied DPO to fine-tune ESM-IF to design more stable proteins. More comparisons are desired.
- The modified DPO algorithm is mostly based on vanilla DPO and is similar to ProteinDPO.
- In Table 1, the DPO fine-tuned ProteinMPNN does not achieve better diversity than vanilla ProteinMPNN significantly.
- In Table 2, even with fine-tuning, ProteinMPNN + DPO is worse than ESM-IF in TM-Score. Therefore, its interesting to fine-tune ESM-IF with the same algorithm.

[1] Aligning protein generative models with experimental fitness via Direct Preference Optimization

**Questions:**

See Weaknesses

---

> ### Author Response · Authors · 2024-11-27
> **Response to Reviewer sGWi**
>
> Dear Reviewer,
>
> Thank you very much for your thoughtful suggestions and comments! We appreciate your patience, and thank you in advance for considering our responses.
>
> > “ProteinDPO [1] has applied DPO to fine-tune ESM-IF to design more stable proteins. More comparisons are desired.”
>
> While ProteinDPO is standard DPO [1] applied to maximize the stability of protein sequences, it does not target the diversity aspect of this design problem. In contrast, our paper proposes a new method specifically designed to jointly maximize sequence quality (TM-Score) and diversity.
>
> Additionally, our paper applies our DPO methods to ProteinMPNN via an OpenFold dataset, whereas ProteinDPO applies DPO to ESM-IF via a stability dataset. In our experiments, ESM-IF is generally a stronger base model, making the comparison unfair. Furthermore, the core algorithm presented in the ProteinDPO paper is standard DPO, plus a weighted variant that performs similarly. We report standard DPO results in our experiments, showing our methods outperform it.
>
> > The modified DPO algorithm is mostly based on vanilla DPO and is similar to ProteinDPO.
>
> We present two DPO modifications: diversity-regularized DPO (Section 3.1), and reward-weighted DPO (Section 3.2). We consider diversity-regularized DPO to be the more novel contribution. While it is based on standard DPO, we argue that it is a significant contribution due to 1) the novel application of post-training alignment to diversity optimization, and 2) the principled derivation/explanation of the online DPO algorithm.
>
> Firstly, we note that DPO-tuned ProteinMPNN has 7-14% worse diversity compared to vanilla ProteinMPNN in our benchmarks (Tables 1-2). This motivated us to create new DPO methods to directly handle diversity. To the author’s knowledge, we are the first work to propose directly optimizing for diversity via DPO or any post-training alignment methods.
>
> Secondly, recent research in the DPO alignment space ([2], [3], [4]) has begun to outline the effectiveness of online sampling in DPO for increasing downstream rewards. To the author’s knowledge, ours is the first work to produce an online DPO method by a simple derivation from the KL-constrained RL objective. In the computational biology space, ours is also the first work to even consider online DPO for protein work (ProteinDPO and [5] use strictly offline methods, i.e. the train dataset remains static throughout training). The comparison between online and offline DPO methods can be seen in Tables 1 and 2, where DPO with diversity regularization (non-zero alpha) matches the TM-score of offline DPO but with 11-18% improvement in diversity. Furthermore, we provide a principled derivation for the online term, whereas prior work in NLP generally applies online sampling ad-hoc, without formal justification [2], [3]. We also provide an empirically supported theory (see the differential entropy formulation) on why online DPO is so helpful compared to offline DPO.
>
> > In Table 1, the DPO fine-tuned ProteinMPNN does not achieve better diversity than vanilla ProteinMPNN significantly.
>
> Compared to vanilla ProteinMPNN, standard DPO finetuning produces a model with 14% worse diversity, but also 8% higher TM-Score (Table 1). With our method (diversity-regularized DPO), we are able to achieve similar diversity as vanilla ProteinMPNN but also maintain that 8% increase in TM-Score. Thus, our DPO method allows us to reach a Pareto improvement over vanilla ProteinMPNN, achieving similar diversity but also better TM-Score. This is a testament to our original goal of jointly optimizing both diversity and TM-Score, rather than having to sacrifice one for the other as standard DPO requires.
>
> > In Table 2, even with fine-tuning, ProteinMPNN + DPO is worse than ESM-IF in TM-Score. Therefore, its interesting to fine-tune ESM-IF with the same algorithm.
>
> We agree very much with the reviewer. Our goal is to show that our DPO method is ultimately base-model agnostic and can lift the performance of ESM-IF, PiFold, etc. in addition to ProteinMPNN. Unfortunately, due to compute constraints and severe logistical limitations with running new experiments, we were regretfully unable to finetune ESM-IF. We hope to see this in future research!
>
> Thank you again for your patience with our responses! We hope this clarifies your concerns and look forward to additional discussion.
>
> [1] Direct Preference Optimization: Your Language Model is Secretly a Reward Model
>
> [2] Bootstrapping Language Models with DPO Implicit Rewards
>
> [3] Direct Language Model Alignment from Online AI Feedback
>
> [4] Understanding the performance gap between online and offline alignment algorithms
>
> [5] Preference optimization of protein language models as a multi-objective binder design paradigm

---

### Meta-Review · Area_Chair_iYDe · 2024-12-21

**Metareview:**

The paper applies Direct Preference Optimization (DPO) to fine-tune the inverse-folding method ProteinMPNN (a message-passing encoder-decoder model), for peptide design. A modified DPO objective is introduced to incorporate a diversity promoting penalty and to leverage domain-specific priors. The approach is demonstrated on popular benchmarks datasets.

The paper is a pleasure to read and the proposed approach is elegant and well-motivated. The connection between diversity and differential entropy is very interesting.

Overall the AC and reviewers all agree that the methodology is promising, but unfortunately several concerns remain in terms of the evaluation to convincingly demonstrate the value and generality of the approach. In particular it is essential to evaluate how general the applicability and benefits of the proposed approach is, by testing against more recent approaches, and by evaluating the benefit of the authors 'versions of DPO on other inverse folding approaches beyond ProteinMPNN wherever possible. It would also important to  bootstrap the computation of TM-score and report self-consistency RMSD for a more complete assessment.

It is unfortunate that the authors encountered data access issues and could not perform additional experiments during rebuttal. We strongly encourage them to do so and resubmit their work.

**Additional Comments On Reviewer Discussion:**

While all reviewers agree that online DPO for diversity optimization is a promising direction for peptide inverse folding, they raised several concerns regarding lack of recent comparison approaches, and limited improvement from the proposed approach.

Due to data access issues, the authors were unfortunately unable to perform additional experiments. Hence their responses, while very valuable, were limited to qualitative comments. As a result the reviewers could not raise their scores significantly as important experiments are lacking.

---

### Decision · Program_Chairs · 2025-01-22

Reject